# Adenosine diphosphate restricts the protein remodeling activity of the Hsp104 chaperone to Hsp70 assisted disaggregation

Agnieszka Kłosowska*, Tomasz Chamera, Krzysztof Liberek*

Department of Molecular and Cellular Biology, Intercollegiate Faculty of Biotechnology, University of Gdańsk and the Medical University of Gdańsk, Gdańsk, Poland

**Abstract** Hsp104 disaggregase provides thermotolerance in yeast by recovering proteins from aggregates in cooperation with the Hsp70 chaperone. Protein disaggregation involves polypeptide extraction from aggregates and its translocation through the central channel of the Hsp104 hexamer. This process relies on adenosine triphosphate (ATP) hydrolysis. Considering that Hsp104 is characterized by low affinity towards ATP and is strongly inhibited by adenosine diphosphate (ADP), we asked how Hsp104 functions at the physiological levels of adenine nucleotides. We demonstrate that physiological levels of ADP highly limit Hsp104 activity. This inhibition, however, is moderated by the Hsp70 chaperone, which allows efficient disaggregation by supporting Hsp104 binding to aggregates but not to non-aggregated, disordered protein substrates. Our results point to an additional level of Hsp104 regulation by Hsp70, which restricts the potentially toxic protein unfolding activity of Hsp104 to the disaggregation process, providing the yeast protein-recovery system with substrate specificity and efficiency in ATP consumption.

*For correspondence: agnieszka. jurczyk@biotech.ug.edu.pl (AK); liberek@biotech.ug.edu.pl (KL)

Competing interests: The authors declare that no competing interests exist.

## Introduction

Molecular chaperones maintain protein homeostasis by supporting protein folding and preventing aggregation. However, during severe stress the capacity of this protective system becomes exhausted due to appearance of excess amounts of misfolded proteins and their aggregation. Once physiological conditions are restored, the aggregation process is reversed by specialized chaperones, disaggregases, capable of reactivation of proteins trapped in aggregates (*Aguado et al., 2015*; *Doyle et al., 2013*; *Kim et al., 2013*; *Mogk et al., 2015*). In yeast *Saccharomyces cerevisiae* this role is played by the Hsp104 chaperone, essential for development of tolerance to stress (*Sanchez and Lindquist, 1990*) and for propagation of prions (*Chernoff et al., 1995*).

Hsp104 is an AAA+ superfamily member (ATPase Associated with various cellular Activities) and its homologs, Hsp104/ClpB proteins, are present in fungi, protozoans, plants and bacteria (*Ammelburg and Frickey, 2006*; *Malinovska et al., 2015*; *Mishra and Grover, 2015*). Hsp104 consists of four domains: the N-terminal domain, two Nucleotide Binding Domains (NBD1 and NBD2) and a coiled coil region protruding from the NBD1, called the M-domain (*Lee et al., 2003*). Hsp104 functions as a hexamer in which the Nucleotide Binding Domains form a double ring (*Carroni et al., 2014*; *Lee et al., 2010*; *Parsell et al., 1994*). Disaggregation is powered by ATP hydrolysis in NBD1 and NBD2 (*Schaupp et al., 2007*; *Wendler et al., 2009*). During disaggregation a polypeptide is disentangled from the aggregate, translocated through the central channel, and enabled to fold into the native structure (*Lum et al., 2004*; *Tessarz et al., 2008*; *Weibezahn et al., 2004*).

**eLife digest** Under stressful conditions, such as high temperatures, many proteins lose their proper structure and clump together to form large irregular aggregates. To combat this effect, living organisms exposed to stress produce specialized proteins called chaperones, which can rescue the damaged proteins from aggregates.

Studies into this "disaggregation" process often use budding yeast as a model organism. The protein-recovery machinery in this yeast is composed of a ring-shaped enzyme called Hsp104, together with a chaperone called Hsp70 and its partner Hsp40. The Hsp104 enzyme converts molecules of ATP into ADP and uses the energy released from the reaction to move, or "translocate", damaged proteins through its central channel and release them from the aggregates.

Previous studies had reported that ADP negatively affects Hsp104. Now, Kłosowska et al show that Hsp104 is almost inactive in a test-tube if the concentration of ADP is as high as that found inside a cell. This raises a question: how can Hsp104 efficiently remove proteins from aggregates in cells if the conditions are so unfavorable?

Using purified proteins, Kłosowska et al. go on to show that Hsp104 is able to tolerate the level of ADP found inside cells thanks to the Hsp70 chaperone. The experiments show that ADP weakens Hsp104's ability to bind proteins while Hsp70 supports this ability and counteracts the negative effect of ADP. Further experiments demonstrate that Hsp104 is less affected by ADP, and binds more readily to ATP, when it is translocating proteins. These findings explain how the yeast disaggregating machinery can work even at relatively high concentrations of ADP, and reveal a new control mechanism in the disaggregation process.

Many important proteins have poorly organized fragments that can be recognized by Hsp104, and if Hsp104 was to bind to and translocate these proteins it could harm the cell. The findings of Kłosowska et al. suggest that Hsp70 helps Hsp104 to specifically bind to and act upon proteins in aggregates, while binding to partly unstructured proteins is limited by the high ADP concentration. Further studies are now needed to understand how the protein-recovery machinery can discriminate between aggregated and non-aggregated proteins.

Hsp104 alone is insufficient for protein recovery from aggregates. A functional disaggregation machinery is composed of Hsp104 cooperating with the Hsp70 chaperone (Ssa1 in *S. cerevisiae*) and its cochaperone Hsp40 (Ydj1 and Sis1 in *S. cerevisiae*) (*Glover and Lindquist, 1998*; *Krzewska et al., 2001*). Hsp70 interacts with the M-domain of Hsp104 and facilitates Hsp104-mediated protein renaturation at several stages of the process (*Miot et al., 2011*; *Rosenzweig et al., 2013*; *Schlee et al., 2004*). Firstly, Hsp70 targets Hsp104 to aggregates and stabilizes the Hsp104-aggregate complexes (*Acebrón et al., 2009*; *Okuda et al., 2015*; *Winkler et al., 2012*). Secondly, Hsp70 activates Hsp104 by affecting the position of the M-domain against the surface of NBD1 (*Haslberger et al., 2007*; *Lee et al., 2013*; *Lipińska et al., 2013*; *Oguchi et al., 2012*; *Seyffer et al., 2012*; *Sielaff and Tsai, 2010*). Finally, Hsp70 might participate in folding events downstream of Hsp104.

The M-domain regulates the activity of the disaggregase. M-domains are assembled around the NBD1 ring in the hexamer. Each M-domain interacts with respective NBD1 via a network of ionic bonds and keeps Hsp104/ClpB proteins in a repressed, disaggregation-compromised state. Hsp70 binding to the M-domain causes its reposition, which transits Hsp104/ClpB into a derepressed state, manifested by highly stimulated ATPase and disaggregation activities (*Oguchi et al., 2012*). The latter state is permanent in the so-called derepressed mutants, which are partially independent from Hsp70 in protein disaggregation and display highly elevated ATPase activity (*Carroni et al., 2014*; *Jackrel et al., 2014*; *Lee et al., 2013*; *Lipińska et al., 2013*; *Oguchi et al., 2012*; *Seyffer et al., 2012*; *Sielaff and Tsai, 2010*). The M-domain mediated control of Hsp104 protein is critical for the cell, as the derepressed mutants are toxic (*Lipińska et al., 2013*; *Schirmer et al., 2004*). The mechanism coupling the M-domain with the increased Hsp104 activity remains not fully understood.

Both Nucleotide Binding Domains of Hsp104 possess typical motifs for an AAA+ protein: Walker A, responsible for ATP binding, Walker B, essential for ATP hydrolysis and arginine fingers,

necessary for the intersubunit communication (*Biter et al., 2012*; *Schaupp et al., 2007*; *Schirmer et al., 2001*; *Zeymer et al., 2014*). ATP hydrolysis and translocation of protein substrates through the central channel are in a strict allosteric interdependence: polypeptide processing stimulates the ATPase activity of Hsp104 (*Tessarz et al., 2008*; *Woo et al., 1992*; *Yamasaki et al., 2015*), while the efficiency of polypeptide binding to Hsp104 is determined by the presence of ATP in NBD1 (*Franzmann et al., 2011*; *Schaupp et al., 2007*). Changes of the nucleotide states of NBDs are associated with domain rearrangements and movements of peptide binding loops located in the central channel. These conformational changes couple the energy generated during ATP hydrolysis with the protein-threading force (*Lum et al., 2004*; *Wendler et al., 2009*).

Remarkably, Hsp104 has low affinity towards ATP and relatively high towards ADP, therefore ADP strongly inhibits ATP hydrolysis (*Franzmann et al., 2011*; *Glover and Lindquist, 1998*; *Grimminger et al., 2004*). For this reason, studies of Hsp104 activity are usually performed under the conditions optimal for its ATPase activity, including saturating (5–10 mM) concentration of ATP and an ATP regeneration system to avoid ADP accumulation. Under such experimental conditions Hsp104 is a potent ATP-hydrolyzing and protein unfolding machine (*Biter et al., 2012*; *Desantis et al., 2014*; *Franzmann et al., 2011*; *Tessarz et al., 2008*). Yet, these nucleotide concentrations are very different from those observed in the cytosol, with ATP at 2.1–3.4 mM concentration and ADP at 0.5–1.5 mM, depending on the growth conditions (*Canelas et al., 2008*; *Teusink et al., 2000*; *Wu et al., 2006*). So far, little has been known about how Hsp104 disaggregase functions under conditions similar to the ones observed in the living cell, where ADP constitutes a significant proportion of adenine nucleotides. Therefore, we addressed the question: how does ADP affect Hsp104 activity?

Here, we report that ADP compromises ATP hydrolysis, substrate binding and translocation by Hsp104 in the absence of the Hsp70 chaperone. We also demonstrate that protein disaggregation at physiological concentrations of ATP and ADP is only effective due to the Hsp70 assistance. Hsp70 promotes Hsp104 binding to the aggregated protein substrate but does not support processing of non-aggregated proteins. Furthermore, we show that the process of polypeptide threading by Hsp104 facilitates binding of ATP, strongly stimulating the ATPase activity of the disaggregase in the presence of ADP. Based on these findings we propose a new level of regulation of the disaggregase by its chaperone partner Hsp70.

## Results

### ADP inhibits the ATPase activity and polypeptide processing by Hsp104

All known Hsp104 activities depend on ATP hydrolysis. In a rough comparison, Hsp104 binds ATP with at least an order of magnitude lower affinity than ADP (*Grimminger et al., 2004*). In the light of the ATP and ADP binding properties we asked how ADP influences Hsp104 functioning. At the saturating ATP concentration (10 mM), Hsp104 catalyzed the hydrolysis of 90 molecules of ATP per minute (*Figure 1A*). However, addition of only 1 mM ADP to the reaction mixture reduced the rate of ATP hydrolysis to 30% and under the ATP:ADP ratio 10:4 the ATPase activity dropped to less than 2% (*Figure 1A*). This illustrates how strong the effect of ADP on Hsp104 activity is, with ATP hydrolysis considerably limited even at the saturating ATP concentration.

Next, we tested the influence of ADP on the Hsp104 activity at the physiological ATP concentration. Recent measurements in intact yeast cells show that ATP concentration oscillates around 2.6 mM (*Ozalp et al., 2010*), which is an intermediate value of the previously reported ATP levels measured in cell extracts (2.1–3.4 mM) (*Canelas et al., 2008*; *Teusink et al., 2000*; *Wu et al., 2006*). This ATP concentration will be referred to as physiological thorough this work, although it needs to be kept in mind that the cellular level of ATP changes when yeast metabolism adapts to the growth conditions (*Larsson et al., 1997*; *Osorio et al., 2003*; *Ozalp et al., 2010*). We measured the ATPase activity of Hsp104 at fixed 2.6 mM ATP and increasing ADP concentrations. At the physiological ATP level the rate of ATP hydrolysis was reduced to only 20 molecules of ATP hydrolyzed per minute per Hsp104 monomer (*Figure 1B*) as compared to 90 ATP molecules at the saturating ATP concentration (*Figure 1A*). ADP in the reaction mixtures caused strong inhibition of the ATPase activity, which dropped to less than 2 ATP molecules per minute at 1 mM ADP. This level of ADP represents

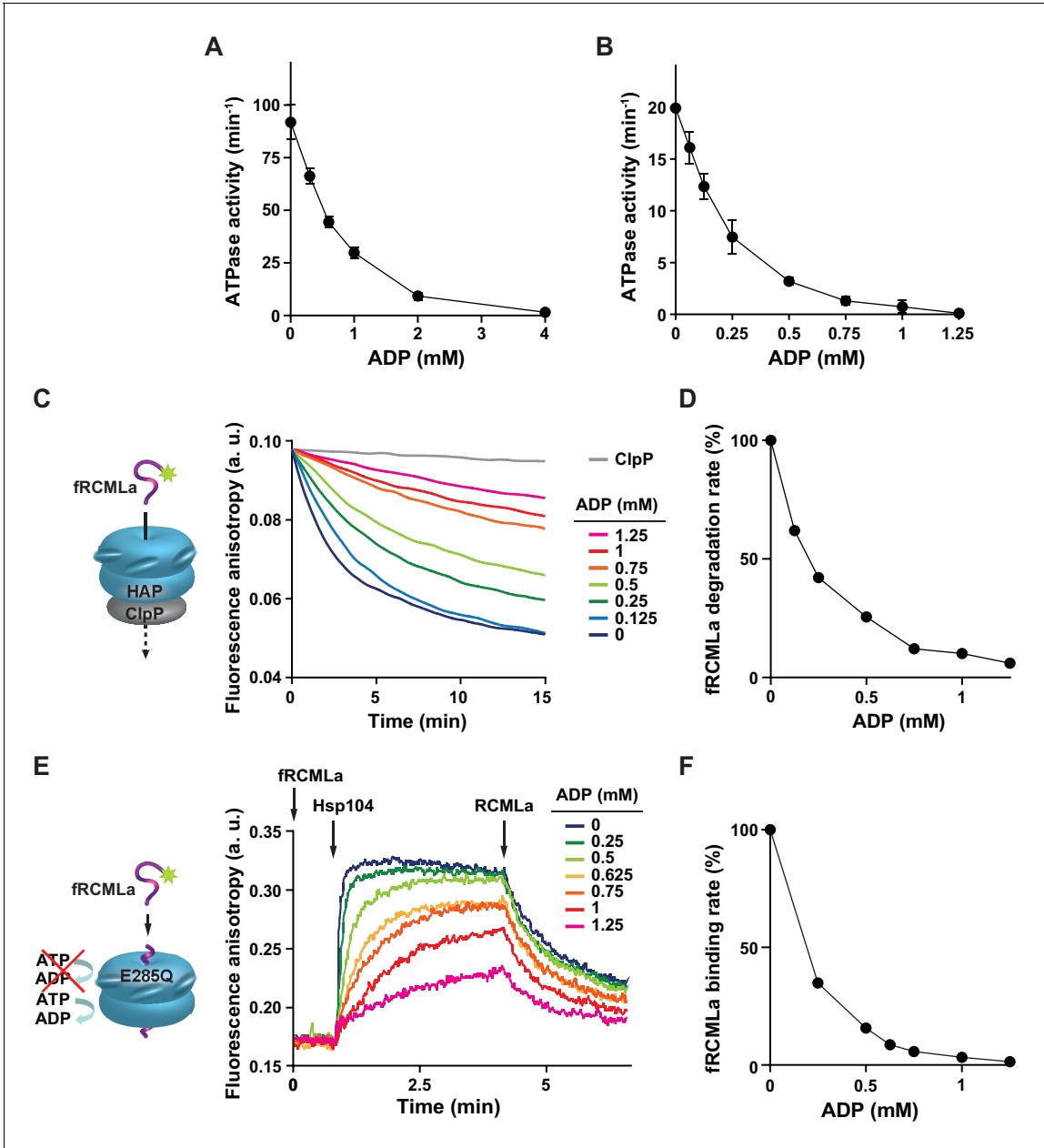

**Figure 1.** ADP restricts Hsp104 activities. (**A**,**B**) ADP strongly inhibits the ATPase activity of Hsp104. The rate of ATP hydrolysis by Hsp104 was assessed (**A**) at 10 mM ATP or (**B**) at 2.6 mM ATP and at the indicated concentrations of ADP. Data are the mean of three independent experiments (± SD). (**C**) ADP inhibits fRCMLa translocation and proteolysis by HAP-ClpP. fRCMLa (5 μM) was incubated at 2.6 mM ATP with HAP (1 μM) and ClpP (1.8 μM) at the indicated concentrations of ADP and its proteolysis was measured by following changes in fluorescence anisotropy. In a control fRCMLa was incubated with ClpP without HAP (grey). (**D**) The rates of fRCMLa proteolysis by HAP-ClpP were calculated from the slopes of the fluorescence anisotropy curves for each ADP concentration shown in (**C**) and normalized to the HAP activity in the absence of ADP. (**E**) ADP impairs binding of Hsp104 to fRCMLa. Hsp104 E285Q (12 μM) was injected to the reaction mixture containing fRCMLa (1 μM), at 2.6 mM ATP and at the ADP concentrations indicated in the legend. After 200 s, non-labeled RCMLa was added to the final concentration of 40 μM. (**F**) The relative initial rates of fRCMLa binding by Hsp104 E285Q at the indicated ADP concentrations were calculated basing on the fluorescence anisotropy curves. a. u. – arbitrary units.

The following figure supplements are available for figure 1:

**Figure supplement 1.** ADP inhibition of protein translocation through HAP.

**Figure supplement 2.** ADP effect on ATP hydrolysis by Hsp104 Walker B mutants.

the average of the ADP concentrations in yeast cells established in the previous studies (0.5–1.5 mM) (*Canelas et al., 2008*; *Teusink et al., 2000*; *Wu et al., 2006*). The degree of inhibition showed similar dependence on ATP:ADP ratio for both ATP levels: 2.6 and 10 mM (*Figure 1A,B*). All the above results show that ADP level is a significant factor determining the ATPase activity of Hsp104.

Considering the strong effect of ADP on Hsp104 ATPase activity we asked how ATP and ADP at the physiological concentrations affect its protein translocation activity. Hsp104 in the absence of the cooperating Hsp70 chaperone is able to bind and translocate proteins which adopt disordered conformation but are not aggregated, such as reduced carboxymethylated lactalbumine (RCMLa) (*Bösl et al., 2005*). We labeled RCMLa with fluorescein (fRCMLa) and employed it to monitor its translocation by Hsp104 in real time. We used Hsp104 variant termed HAP which was modified to interact with the bacterial protease ClpP so that every polypeptide that has been translocated through the central channel of HAP undergoes proteolysis (*Tessarz et al., 2008*). When fRCMLa was incubated with HAP and ClpP at 2.6 mM ATP, we observed a decrease in fluorescence anisotropy, corresponding to fRCMLa degradation and release of short fluorescein-labeled peptides (*Figure 1C*). We plotted the fRCMLa proteolysis rate against ADP concentrations (*Figure 1D*). With increasing level of ADP fRCMLa proteolysis by HAP-ClpP slowed down, with a 10-fold reduction of the activity at 1 mM ADP. Thus, under the physiological proportions of adenine nucleotides not only ATP hydrolysis but also translocation of disordered proteins by Hsp104 is impaired. The effect of ADP was similar in an analogous experiments carried out at 10 mM ATP and at different fRCMLa concentrations (*Figure 1—figure supplement 1A,B*), showing that the degree of inhibition depends predominantly on ATP:ADP ratio.

Next, we asked whether the strong inhibition of substrate processing by Hsp104 was associated with a defect in protein binding. We used an Hsp104 E285Q variant in the binding experiments because a complex between this mutant and the substrate is much more stable compared to the WT Hsp104 (*Franzmann et al., 2011*). The E285Q mutation in the Walker B motif affects NBD1 in such way that this domain binds ATP but is not able to hydrolyze it (*Schaupp et al., 2007*). As a result, NBD2 remains the only hydrolytically active domain of Hsp104. We chose Hsp104 E285Q because its ATPase activity was inhibited by ADP to the same degree as of the WT protein, while Hsp104 with an analogous Walker B substitution in NBD2 (E687Q) showed much lower degree of inhibition under the same conditions (*Figure 1—figure supplement 2*). We used fluorescence anisotropy to monitor Hsp104 E285Q interaction with fRCMLa. We added Hsp104 E285Q to fRCMLa

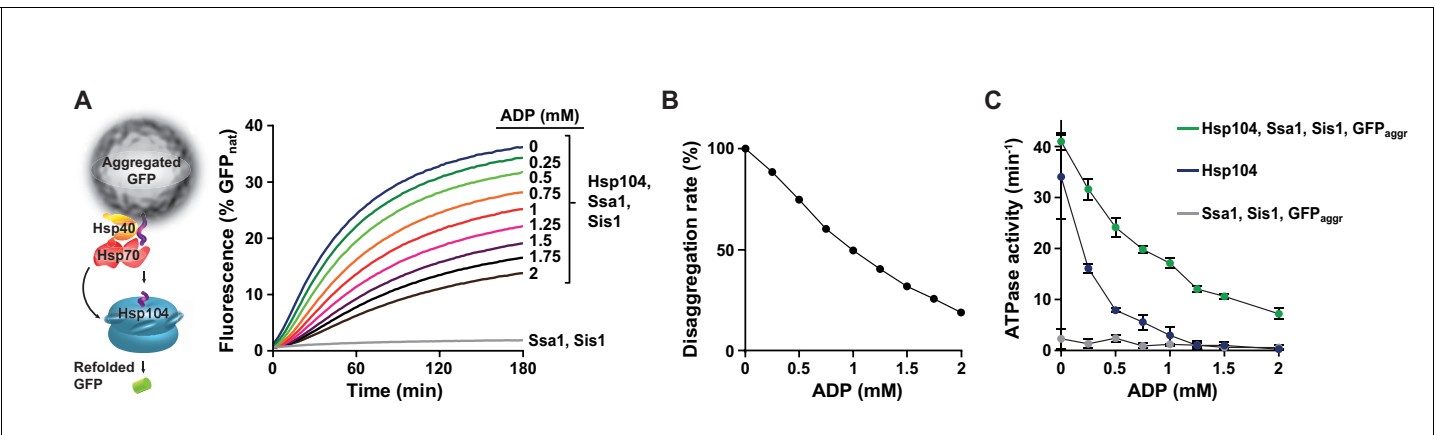

**Figure 2.** Hsp70 allows Hsp104 to overcome ADP inhibition. (**A**) ADP impact on GFP disaggregation by Hsp104 and the Hsp70 system. Heat-aggregated GFP (0.04 mg ml⁻¹) was incubated with Hsp104 (1 µM), Ssa1 (2 µM) and Sis1 (0.4 µM) at 2.6 mM ATP and at the indicated concentrations of ADP. A control experiment was performed without Hsp104 (grey). (**B**) The relative GFP renaturation rates by Hsp104 and Hsp70-Hsp40 at the indicated ADP concentrations were calculated from the initial slope of the fluorescence curves from (**A**). (**C**) ADP effect on ATP hydrolysis by Hsp104 during the Hsp70-assisted disaggregation. ATPase activity was measured for Hsp104 (1 µM) in the presence of Ssa1 (1 µM), Sis1 (0.1 µM) and aggregated GFP (0.2 mg ml⁻¹) (green) or for Hsp104 alone (blue) under the conditions as in (**A**). For comparison, the ATPase activity of Ssa1, with Sis1 and GFP and without Hsp104 was assessed under the same conditions (grey). Data are the average of three experiments (± SD).

at constant 2.6 mM ATP and changing ADP concentrations and recorded an increase in the anisotropy signal over time, reflecting binding of Hsp104 to fRCMLa (*Figure 1E*). We plotted the binding rate against ADP concentration (*Figure 1F*). At 2.6 mM ATP we observed fast binding of fRCMLa but with increasing ADP the rate of fRCMLa binding decreased, reaching 3% at 1 mM ADP. Addition of an excess of unlabeled RCMLa to the Hsp104 E285Q-fRCMLa complex caused a drop in fluorescence anisotropy, as the fluorescently labeled substrate was released from Hsp104 (*Figure 1E*).

All the above results show that at the physiological concentrations of adenine nucleotides the biochemical activities of Hsp104 such as ATP hydrolysis, substrate binding and translocation through the central channel are very strongly inhibited.

## Hsp70 allows Hsp104 to overcome ADP inhibition

Hsp104 provides thermotolerance in vivo at the physiological concentrations of ATP and ADP, which implies that Hsp104 is active in disaggregation regardless of the disadvantageous nucleotide ratio (*Sanchez and Lindquist, 1990*). The recovery of aggregated proteins, however, contrary to the processing of disordered proteins, requires collaboration with the Hsp70-Hsp40 machinery. Therefore, we asked how the physiological concentrations of adenine nucleotides affect protein disaggregation performed by Hsp104 in cooperation with the Hsp70 chaperone. As a substrate we used heat-aggregated GFP, which does not require Hsp70 for folding downstream of Hsp104. We incubated aggregated GFP with Hsp104 in the presence of Hsp70 (Ssa1) and its co-chaperone Hsp40 (Sis1) at constant 2.6 mM ATP and changing ADP concentrations and monitored the recovery of GFP fluorescence (*Figure 2A*). We plotted the initial GFP folding rate against ADP concentration (*Figure 2B*). Strikingly, at 1 mM ADP the rate of disaggregation and refolding of GFP was reduced only by 50%. The inhibitory effect of ADP on protein disaggregation by the Hsp104-Hsp70 bi-chaperone system

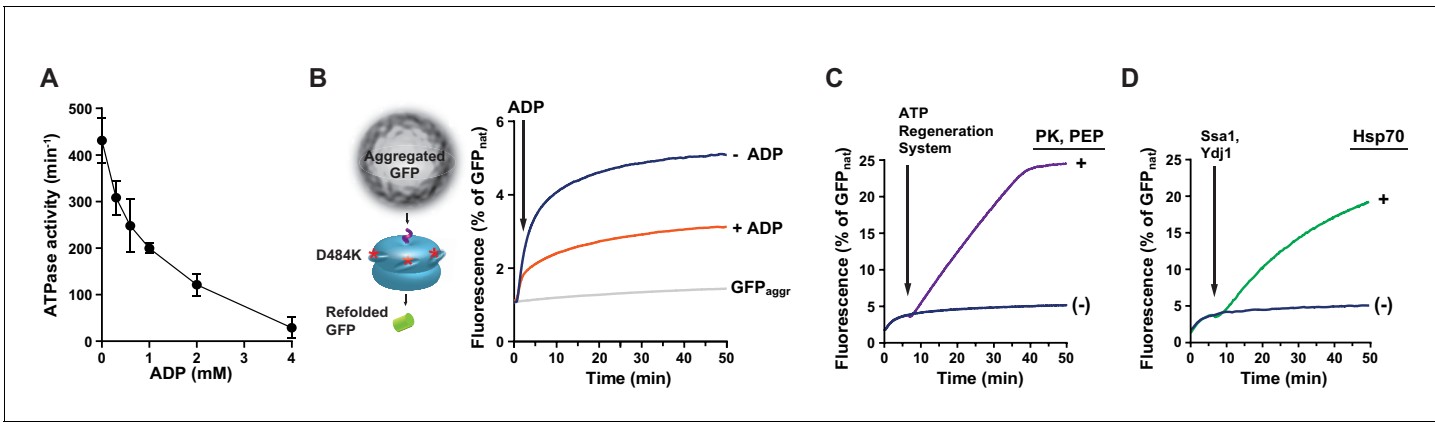

**Figure 3.** The derepressed D484K variant of Hsp104 is inhibited by ADP. (**A**) The ATPase activity of Hsp104 D484K is strongly affected by ADP. ATPase activity of D484K variant was measured at 10 mM ATP and at the indicated concentrations of ADP. Values are the average of three independent experiments (± SD). (**B**) ADP inhibits the disaggregation activity of Hsp104 D484K in the absence of Hsp70. Disaggregation of heat-aggregated GFP (0.04 mg ml⁻¹) by Hsp104 D484K (0.5 μM) at 10 mM ATP (blue). After 60 s of the reaction, ADP was added to 2 mM concentration (red). (**C**) ATP regeneration system or (**D**) The Hsp70 system restores the disaggregation activity of Hsp104 D484K. The experiment was initiated as in (**B**), and after 5 min (**C**) an ATP regeneration system comprising PK (0.1 mg ml⁻¹) and PEP (40 mM) (purple) or (**D**) the Hsp70 chaperone system: Ssa1 (2 μM) and Ydj1 (0.5 μM) (green) was added to the reaction mixture.

The following figure supplements are available for figure 3:

**Figure supplement 1.** Hsp104 D484K is affected by ADP similarly as the WT Hsp104.

**Figure supplement 2.** Protein translocation activity of HAP D484K is inhibited by ADP.

**Figure supplement 3.** Hsp70 allows efficient disaggregation at low ATP:ADP ratio.

**Figure supplement 4.** In the absence of ADP the derepressed Hsp104 D484K is independent of Hsp70 in disaggregation.

was therefore much milder than the 90% decrease in the protein translocation activity and the 97% reduction in the substrate binding rate observed for Hsp104 alone under the same conditions (*Figure 1D,F*). This shows that Hsp104 response to ADP is different in the processes dependent on and independent from the Hsp70 chaperone.

Since protein disaggregation is powered by ATP hydrolysis, we anticipated that the ATPase activity of Hsp104 would also be less affected by ADP during the recovery of aggregated proteins. To verify this, we measured the rate of ATP hydrolysis in the presence of the Hsp70-Hsp40 system (Ssa1, Sis1) and aggregated GFP (*Figure 2C*). We halved the concentration of Hsp70, an ATP consuming protein, to minimize its contribution to ADP production. Under these conditions the ATPase activity of Hsp104 was less sensitive to ADP, similarly as observed in the GFP disaggregation assay (*Figure 2B*). For example, at the ATP:ADP ratio 2.6:1 the rate of ATP hydrolysis was reduced by 50%, whereas for Hsp104 alone the ATPase activity was inhibited by 90% (*Figure 2C*). Together, these results show that Hsp70 moderates Hsp104 response to ADP.

One possible explanation is that Hsp70 releases Hsp104 from ADP inhibition because it interacts with and repositions the M-domain, which abrogates Hsp104 repression and results in the elevated ATPase activity (*Oguchi et al., 2012*). To test this hypothesis we used a derepressed variant of Hsp104, D484K, in which an ionic interaction between NBD1 and the M-domain is disrupted (*Lipińska et al., 2013*). Hsp104 D484K displayed over five times higher rate of ATP hydrolysis than WT Hsp104, while the affinity towards ATP was comparable for both protein variants (the *apparent* $K_M$ values: 4.2 mM for WT and 4.7 mM for D484K). Moreover, the ATPase activity of the derepressed D484K was strongly reduced under the unfavorable ATP:ADP ratio (*Figure 3A*), similarly as observed for the WT (*Figure 1A*), regardless of ATP concentration (*Figure 3—figure supplement 1*).

We also measured ADP impact on the protein translocation activity of Hsp104 D484K. We monitored fRCMLa proteolysis by HAP D484K and ClpP at 10 mM ATP. To compensate for the rapid ADP production by HAP D484K, we included an ATP regeneration system, which increased the rate of fRCMLa degradation three times (*Figure 3—figure supplement 2*). At 2 mM ADP a 5-fold inhibition of HAP D484K activity was observed (*Figure 3—figure supplement 2*). It is worth noticing, that although the proteolysis rate was much higher for the derepressed D484K variant than for HAP lacking the D484K substitution (HAP WT), a similar degree of inhibition by ADP was observed for both proteins (*Figure 1—figure supplement 1A,B*, *Figure 3—figure supplement 2*).

Our results suggest that the reposition of the M-domain, even though stimulating, does not overcome the ADP-dependent inhibition of the ATPase activity and the protein processing activity of Hsp104.

Using Hsp104 D484K variant allowed us to analyze reactivation of aggregated proteins independently of Hsp70 (*Lipińska et al., 2013*). To assess the influence of ADP on this process, we incubated heat-aggregated GFP with Hsp104 D484K at 10 mM ATP. Disaggregation of GFP by Hsp104 D484K proceeded efficiently for only approximately 5 min (*Figure 3B*). The analysis of ADP concentration in the disaggregation reaction mixture showed that GFP reactivation stopped when ADP exceeded 2 mM concentration (*Figure 3—figure supplement 3A*). Accordingly, addition of 2 mM ADP after the first minute of the reaction stopped GFP reactivation almost immediately (*Figure 3B*). These results show that ADP strongly inhibits disaggregation performed by the derepressed Hsp104 variant in the absence of Hsp70. In agreement with this, when the accumulated ADP was removed from the reaction by an ATP regeneration system, GFP reactivation resumed and proceeded at high rate (*Figure 3C*). This shows that the derepressed Hsp104 variant efficiently disaggregate proteins from aggregates on its own, as long as it is not inhibited by ADP.

Knowing that Hsp70 enables protein reactivation by the WT Hsp104 in the presence of ADP (*Figure 2A*), we analyzed the influence of Hsp70 on GFP disaggregation by the Hsp104 D484K variant. In the absence of Hsp70, the recovery of GFP fluorescence stopped almost completely after approximately 5 min of disaggregation. When Hsp70-Hsp40 chaperones were added at this time, GFP reactivation resumed and proceeded efficiently (*Figure 3D*). In a similar experiment we monitored ADP concentration during GFP reactivation (*Figure 3—figure supplement 3B*). Hsp70-Hsp40 chaperones allowed Hsp104 to disaggregate GFP in spite of the accumulation of ADP. The reaction proceeded efficiently even at ADP concentration above 2 mM. These results indicate that the Hsp70 chaperone system makes the disaggregation activity of the derepressed Hsp104 D484K variant substantially less sensitive to ADP.

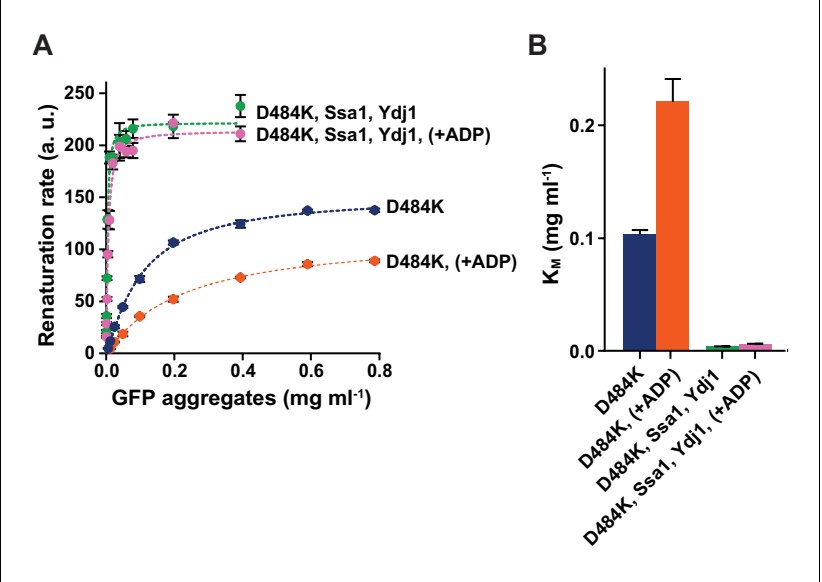

**Figure 4.** Hsp70 supports Hsp104 in disaggregation, compensating the effect of ADP. (**A**) The opposing effects of ADP and Hsp70 on Hsp104 affinity towards aggregates. Disaggregation of heat-aggregated GFP present at different concentrations by Hsp104 D484K variant (0.06 μM) was assessed at 10 mM ATP, with or without ADP (1 mM), in the presence or absence of Ssa1 (2 μM) and Ydj1 (0.5 μM), as indicated in the legend. The initial rate of recovery of GFP fluorescence was plotted against GFP concentration. Data are the means of three measurements (± SD). Dashed lines represent Michaelis-Menten curves fitted to each set of experimental data using least squares fitting with the GraphPrism software. a. u. – arbitrary units. (**B**) Apparent $K_M$ values calculated for each experiment described in (**A**). Values are average of three experiments (± SD).

Further, we asked if the disaggregation activity of the derepressed Hsp104 D484K variant is additionally stimulated by Hsp70 if the inhibitor, ADP, is removed from the reaction. To address this, we compared renaturation of GFP by Hsp104 D484K performed with and without the Hsp70 system and an ATP regeneration system (*Figure 3—figure supplement 4*). When ADP accumulated, Hsp104 D484K was an efficient disaggregase only when its Hsp70 partner was present. However, when ADP was being removed, Hsp104 D484K alone was highly effective in disaggregation and addition of Hsp70 barely stimulated its activity. This indicates that Hsp104 D484K is susceptible to strong inhibition by ADP and only in this context the derepressed disaggregase is Hsp70-dependent.

Our results show that the release of the M-domain mediated Hsp104 repression is insufficient for Hsp104 to tolerate ADP. This raises a question, which other function of Hsp70 accounts for overcoming ADP inhibition of protein disaggregation by Hsp104.

## Hsp70 supports Hsp104 binding to aggregates but not to disordered proteins

Hsp70 not only facilitates disaggregation by the M-domain driven derepression of Hsp104 but also acts in the phase preceding translocation of the substrate by Hsp104. The upstream role of Hsp70 in disaggregation involves initial aggregate remodeling (*Zietkiewicz et al., 2006*) and targeting Hsp104 to aggregates (*Acebrón et al., 2009*; *Okuda et al., 2015*; *Seyffer et al., 2012*). Therefore, we asked whether the latter function of Hsp70 enables disaggregation in the presence of ADP. To measure the upstream effect of Hsp70 we monitored GFP refolding by the derepressed Hsp104 D484K variant in the presence or absence of the Hsp70 chaperone system (Ssa1 and Ydj1) at the increasing levels of GFP aggregates (*Figure 4A*). We fitted the Michaelis-Menten curve to the disaggregation rates plotted versus GFP concentrations and calculated the apparent $K_M$ values to evaluate Hsp104 affinities for the aggregated substrate. Strikingly, Hsp70 increased Hsp104 affinity towards GFP over 25 times (*Figure 4B*). When we compared the influence of ADP and Hsp70 on Hsp104 affinity towards aggregates, the effect of Hsp70 was sufficient to compensate for the two-

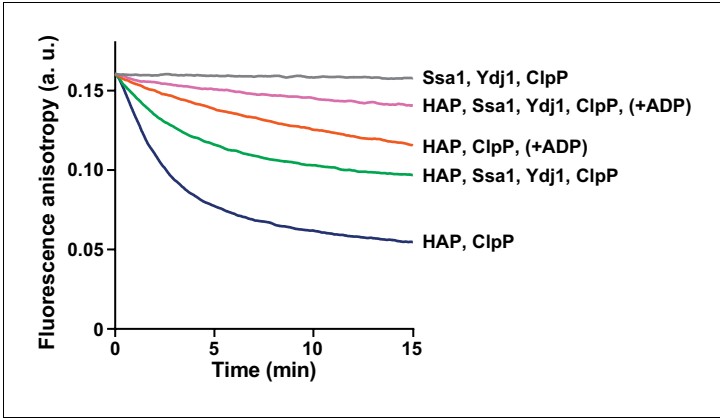

**Figure 5.** Hsp70 does not support Hsp104 in processing of disordered, non-aggregated proteins. Proteolysis of fRCMLa (5 µM) by HAP (1 µM) and ClpP (1.8 µM), carried out at 2.6 mM ATP with or without 1 mM ADP and in the presence or absence of Ssa1 (2 µM) and Ydj1 (0.5 µM), as indicated. Grey line shows a control experiment, in which HAP was omitted. a. u. – arbitrary units.

The following figure supplements are available for figure 5:

**Figure supplement 1.** Hsp70 interaction with fRCMLa.

**Figure supplement 2.** Hsp70 does not support Hsp104 in processing of α-casein.

**Figure supplement 3.** Derepressed HAP D484K is efficient in translocation of disordered proteins at the physiological ATP and ADP concentrations.

---

fold increase in the *apparent* $K_M$ caused by ADP (*Figure 4B*). This result suggests that Hsp70-dependent support in targeting of Hsp104 to aggregates allows overcoming ADP inhibition of protein substrate binding.

Since Hsp70 recruits Hsp104 to aggregates, we asked whether it also promotes Hsp104 binding to unfolded, but non-aggregated proteins, such as RCMLa. As reported previously, Hsp70 protein interacts with RCMLa (*Bimston et al., 1998*). In accordance, an addition of Hsp70-Hsp40 to fRCMLa resulted in a sharp increase in the fluorescence anisotropy signal, indicating that Hsp70-fRCMLa complex is formed (*Figure 5—figure supplement 1*). Next, we examined the Hsp70 and ADP influence on fRCMLa processing by Hsp104 HAP variant accompanied by ClpP. Addition of Hsp70 did not promote fRCMLa translocation and degradation (*Figure 5*). Moreover, ADP inhibited the degradation of fRCMLa regardless of the presence of Hsp70 (*Figure 5*). Interestingly, addition of Hsp70 did not result in increased fRCMLa proteolysis but inhibition of this process was observed. Since both Hsp104 and Hsp70 are able to bind to the same substrate, the observed inhibition can be explained by competition between Hsp70 and Hsp104 for binding to fRCMLa.

To substantiate our results, we assessed Hsp70 influence on HAP-dependent processing of another intrinsically disordered model substrate, α-casein. Hsp70 inhibited its translocation and proteolysis by HAP-ClpP in an analogous way as in case of fRCMLa (*Figure 5—figure supplement 2*).

The above experiments suggest that while Hsp70 significantly promotes binding of Hsp104 to aggregates, it does not facilitate binding of disordered, non-aggregated protein substrates by the disaggregase.

## Protein processing by Hsp104 alleviates inhibition by ADP

Protein reactivation requires ATP hydrolysis by Hsp104, an activity that is severely impaired at the physiological concentrations of adenine nucleotides (*Figure 1B*). However, the impact of ADP on the ATPase activity of Hsp104 becomes weaker during protein reactivation by Hsp104-Hsp70 chaperones (*Figure 2C*). Considering that: (*i*) Hsp70 acts upstream of Hsp104 (*Figure 4A,B*), initiating substrate binding and translocation through Hsp104, and (*ii*) the protein translocation process is associated with elevated ATPase activity of Hsp104 (*Cashikar et al., 2002*; *Woo et al., 1992*), we

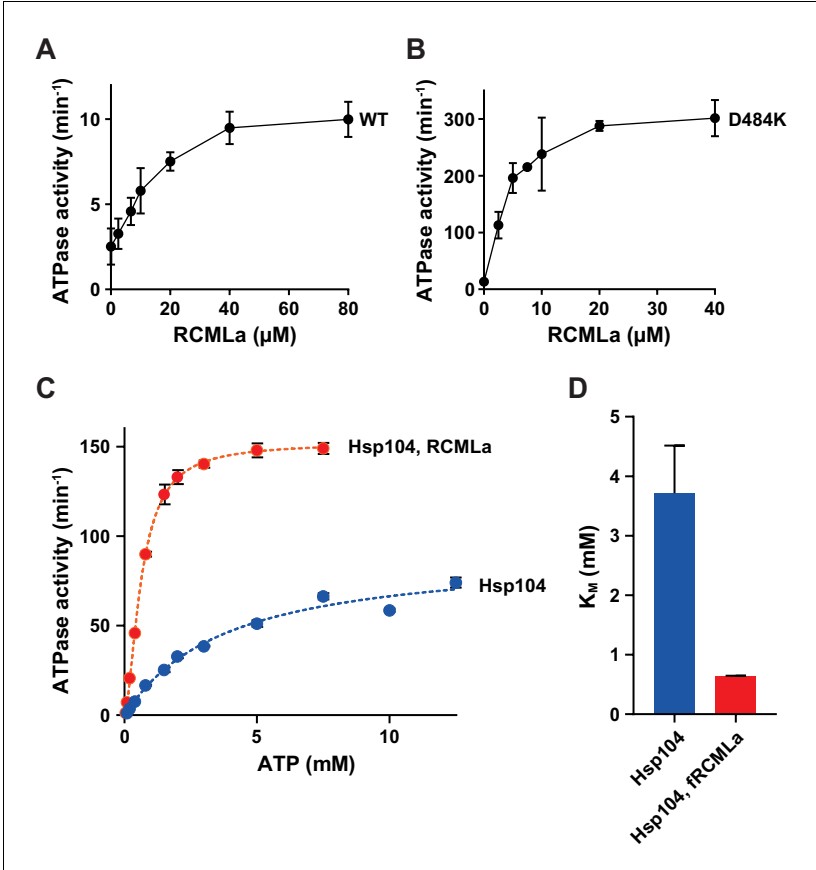

**Figure 6.** ADP inhibition is released during protein translocation by Hsp104. Protein substrate allows efficient ATP hydrolysis at the physiological concentrations of adenine nucleotides. ATPase activity measured for (**A**) Hsp104 or (**B**) Hsp104 D484K variant incubated with different concentrations of RCMLa at 2.6 mM ATP and 1 mM ADP. (**C**) Steady-state ATPase activity of Hsp104 plotted against ATP concentration in the presence (red) or absence (blue) of 50 μM RCMLa. Dashed lines show Michaelis-Menten curves fitted to the experimental data. (**D**) Apparent $K_M$ values calculated for the experiments presented in (**C**). Data represent the mean of three experiments (± SD).

The following figure supplement is available for figure 6:

**Figure supplement 1.** Impact of fRCMLa on ATP hydrolysis by Hsp104 is stronger in the presence of ADP.

speculated that protein translocation by Hsp104 releases ADP-dependent inhibition of its ATPase activity.

To test this hypothesis, we measured the rate of ATP hydrolysis at 2.6 mM ATP and 1 mM ADP, with increasing concentrations of the unfolded substrate, RCMLa (*Figure 6A*). Relatively high concentration of RCMLa was required to compensate for the ADP-dependent inhibition of substrate binding (*Figure 1E,F*). RCMLa at the saturating concentration stimulated the ATPase activity of the ADP inhibited Hsp104 over 5 times. (*Figure 6A*). Even higher stimulation (19-fold) was observed for the derepressed Hsp104 D484K variant (*Figure 6B*). In this case, the ATPase activity reached maximum at lower RCMLa concentration, implying higher affinity of the derepressed Hsp104 for protein substrates (*Figure 6B*). These results indicate that processing of disordered proteins stimulates the ATPase activity of the disaggregase under the, otherwise limiting, physiological concentrations of adenine nucleotides.

It is worth noticing that the positive influence of RCMLa on Hsp104 was more pronounced when ADP was present in the reaction mixture (*Figure 6—figure supplement 1*). Under the physiological ATP and ADP concentrations, the excess of RCMLa stimulated Hsp104 ATPase activity three times more efficiently as compared to the reaction where no ADP was added. This difference was over

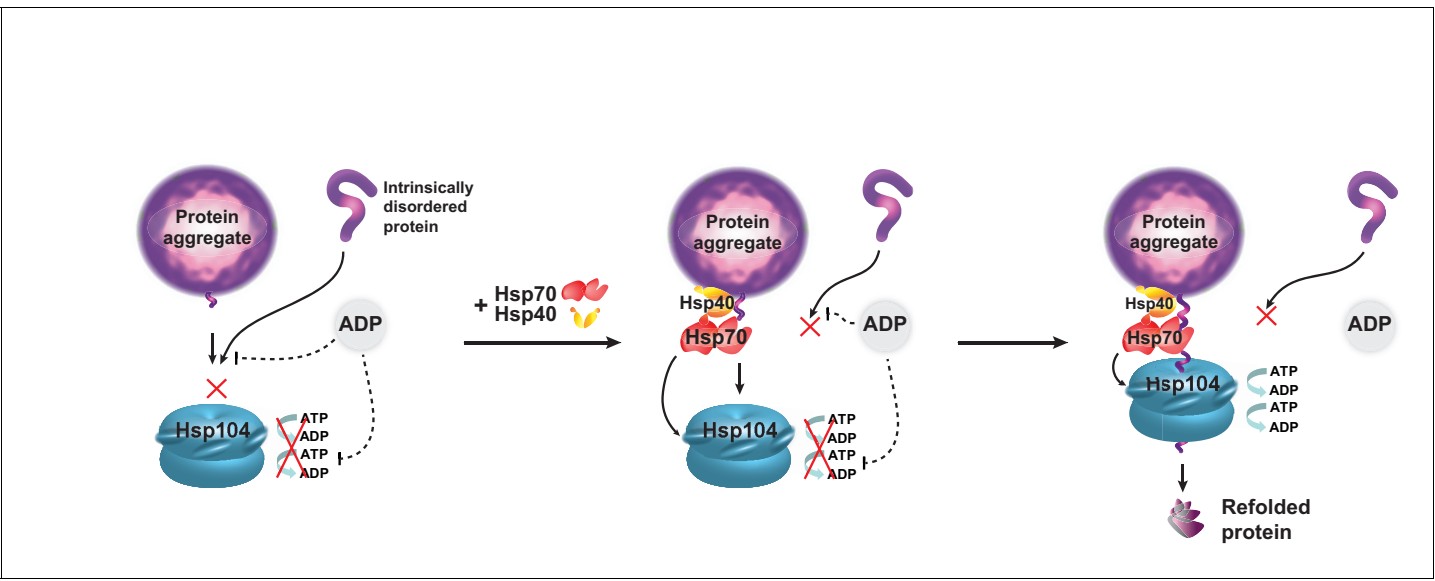

**Figure 7.** The mechanism of Hsp70-mediated Hsp104 activation in the presence of ADP. The cellular level of ADP limits the ATPase activity and restricts Hsp104 interaction with protein substrates. Hsp70 promotes Hsp104 binding to aggregates, but not to intrinsically disordered proteins. When Hsp104 binds a polypeptide and translocation occurs, inhibition by ADP is overcome and the ATPase activity is restored.

two times more pronounced for the derepressed Hsp104 D484K variant. Since ADP competes with ATP for binding to Hsp104, the observed effects suggest that protein translocation could affect Hsp104 affinity towards nucleotides. To verify it, we measured the ATPase activity of Hsp104 at varying ATP concentrations either at the saturating concentration of RCMLa or in the absence of RCMLa (*Figure 6C*). This experimental setup allowed us to compare Hsp104 affinities towards ATP in the presence and absence of the translocated substrate (*Figure 6D*). In the presence of RCMLa, the apparent $K_M$ decreased five times, implying that polypeptide translocation highly increases Hsp104 affinity towards ATP. Together, these results show that translocation of the polypeptides through the Hsp104 central channel facilitates ATP binding to Hsp104 and allows to overcome the ADP-dependent inhibition of its ATPase activity.

Summarized, our results show that Hsp70 allows Hsp104 to overcome the strong inhibitory effect of ADP on Hsp104 disaggregation activity. While ADP negatively affects binding of the protein substrates, Hsp70 substantially increases Hsp104 affinity towards aggregated proteins. At the same time, Hsp70 does not promote processing of disordered, non-aggregated proteins. As a consequence, at the physiological concentrations of adenine nucleotides and in the presence of the Hsp70 chaperone renaturation of aggregated proteins by Hsp104 is efficient but binding and translocation of non-aggregated proteins is strongly inhibited by ADP.

## Discussion

In this work we investigated in vitro Hsp104 functioning at physiological ATP and ADP concentrations. Under such conditions the activities displayed by Hsp104 alone, including ATP hydrolysis as well as binding and translocation of substrate proteins, are highly limited (*Figure 7*). Protein translocation through the central channel is initiated only when Hsp70 specifically recruits Hsp104 for disaggregation by facilitating its interaction with the aggregate. Translocation of the polypeptide substrate leads to the release of ADP inhibition. The release is due to a several-fold increase in Hsp104 affinity towards ATP. Consequently, the process of polypeptide threading highly stimulates the ATPase activity of Hsp104 at non-saturating concentration of ATP and in the presence of the competitive inhibitor, ADP (*Figure 7*). The presented results point to a novel, ADP and Hsp70 dependent, level of regulation of the disaggregase.

The strong inhibition by ADP of Hsp104 in the absence of a protein substrate results from the discrepancy between ATP and ADP binding affinities (*Grimminger et al., 2004*). According to the

previous studies (*Canelas et al., 2008*; *Teusink et al., 2000*; *Wu et al., 2006*), the cellular level of ATP (2.1–3.4 mM) is below saturation for Hsp104 (~10 mM). Furthermore, depending on growth conditions, residual ADP can constitute a significant fraction of adenine nucleotides in the cytosol (up to 1.5 mM) (*Canelas et al., 2008*; *Larsson et al., 1997*; *Osorio et al., 2003*; *Ozalp et al., 2010*). The conditions in yeast cytoplasm are therefore far from optimal for the biochemical activity of the disaggregase. In this work we measured the ATPase and the protein-processing activity of Hsp104 under a broad range ADP concentrations, establishing the relation between the degree of inhibition and ATP:ADP ratio. The highest ATP:ADP ratio reported for yeast cells is 7:1 (*Theobald et al., 1997*). Even at slightly higher ATP:ADP ratio 10:1 the activity displayed by Hsp104 alone was highly reduced (by over 50%) and under the intermediate physiological ATP:ADP ratio 2.6:1 (*Canelas et al., 2008*; *Ozalp et al., 2010*) it was almost completely abolished (inhibited by 90% or more). In contrast, the disaggregation activity of Hsp104 in the presence of Hsp70 at the ATP:ADP ratio 10:1 was reduced by less than 10% and at 2.6:1 - to 50%. 90% degree of inhibition occured below the ATP:ADP ratio 1.3:1, while the lowest proportion of ATP to ADP reported in yeast is 1.4:1 (*Hynne et al., 2001*). Therefore, inhibition by ADP and the counteracting effect of Hsp70 regulate Hsp104 activity across a whole range of physiological proportions between ATP and ADP.

The mechanism of ADP inhibition of Hsp104 can be explained in the light of the previous studies, which show that ADP predominantly occupies the NBD2 domain. It was proposed that ADP release from NBD2 could be a rate-limiting step in the ATPase cycle (*Franzmann et al., 2011*). Agreeably, we observed that ADP strongly inhibits the ATPase activity in NBD2 in the Hsp104 E285Q variant and exerts much milder effect on ATP hydrolysis by Hsp104 E687Q variant, in which NBD1 is the only active ATPase domain. This shows that ADP inhibits Hsp104 by affecting its NBD2.

Interestingly, WT Hsp104, in which both NBDs are functional, responds to ADP similarly as the Hsp104 E285Q variant, in which only the NBD2 domain is hydrolytically active. This suggests that in the WT protein the difficulties in ADP release from NBD2 prevent ATP hydrolysis also in NBD1. It has been established before that when ADP occupies NBD2, NBD1 is mostly in the nucleotide-free state (*Franzmann et al., 2011*). It has also been reported that the absence of nucleotide in NBD1 affects the ATPase activity of the neighbouring NBDs and interrupts protein binding to Hsp104 (*Schaupp et al., 2007*). This could explain how the inhibitory effect of ADP on NBD2 is allosterically transmitted between all the ATPase domains within the hexamer.

Results presented in this work imply an existence of two Hsp104 functional modes, depending on substrate translocation through the central channel: ADP-inhibited and active. In the ADP-inhibited mode both the ATPase activity and polypeptide binding are severely impaired and the protein substrate cannot undergo translocation through Hsp104. In the active mode the presence of the polypeptide in the central channel facilitates ATP binding, and most likely accelerates the rate-limiting step in the ATPase cycle: ADP release. These findings shed new light on how unfolded substrates stimulate ATP hydrolysis by Hsp104, however to fully understand the underlying mechanism detailed structural data collected in the presence of a substrate would be required.

Our data suggest that transition from ADP-inhibited into the active mode occurs upon a successful interaction with a substrate. Although substrate binding is allowed in the presence of ADP, the probability of this event decreases with the increasing proportion of ADP to ATP. It requires high level of protein substrate to compensate for this effect. This underlines the importance of the upstream role of Hsp70 in disaggregation, which involves initial aggregate remodeling and promoting of interaction between Hsp104 and the aggregate. Due to these activities of Hsp70, Hsp104 can bind aggregated substrates, which are at low concentration.

Curiously, the upstream role of Hsp70 is limited to aggregates. Hsp70 does not support binding of non-aggregated protein substrates, allowing Hsp104 activation only at the face of an aggregate. Thus, by specifically targeting Hsp104 towards aggregates, Hsp70 provides another level of control of the disaggregase.

The proposed regulatory mechanism works in addition to the activation of Hsp104 associated with Hsp70-dependent reposition of the M-domain, since the latter effect is not enough to overcome ADP inhibition. This is suggested by the observations that both the wild type Hsp104 and the derepressed D484K mutant of the M-domain are strongly affected by ADP and, in both cases, Hsp70 enables the Hsp104 variants to efficiently disaggregate proteins in the presence of the inhibitor. It needs to be kept in mind, however, that the two roles of Hsp70 are probably interdependent,

as stimulation of the ATPase activity via the M-domain might favor formation of the complex between Hsp104 and an aggregate.

Interestingly, in the absence of Hsp70 addition of an efficient ATP regeneration system is enough to unleash the disaggregating potential of the derepressed Hsp104 D484K. Thus, when Hsp104 is not subjected to the repression by the M-domain, it depends on the Hsp70 chaperone partner exclusively in the context of the release of ADP inhibition.

The restrictive effect of ADP may play an important role in proper and efficient Hsp104 functioning in the cell. Firstly, inhibition of ATP hydrolysis decreases energy dispersing by Hsp104. Secondly, Hsp104 inhibition by ADP prevents a potentially harmful activity of the disaggregase. Such is manifested upon expression of derepressed Hsp104 variants in yeast, which causes disruption of cytoskeleton and inhibits cell division (*Lipińska et al., 2013*; *Schirmer et al., 2004*). This shows that the protein unfolding activity of Hsp104, although highly advantageous under stress conditions, needs to be under strict control, as instead of aggregates it can turn against other protein targets.

Notably, in our in vitro experiments, Hsp70 chaperone supports Hsp104 in processing of aggregates, but not of the model disordered protein substrates. The observation that Hsp70 inhibits both fRCMLa and α-casein proteolysis by the HAP variant of Hsp104 (and ClpP), implies that competition between the chaperones for their binding sites on the unfolded substrates may occur. This points to a plausible shielding role of Hsp70, protecting non-aggregated, intrinsically disordered proteins against the protein-threading activity of Hsp104.

Translocation of the disordered substrates by HAP WT was at very low level at the physiological concentrations of adenine nucleotides in the presence of Hsp70 (*Figure 5*). In comparison, the derepressed HAP D484K variant was still highly active under the same conditions, even upon inhibition by ADP (*Figure 5—figure supplement 3*). The high disaggregation activity of the derepressed Hsp104 comes at the price of high and unspecific protein unfolding activity, which might be responsible for the toxic effects of Hsp104 D484K in the cell (*Lipińska et al., 2013*). Meanwhile, the M-domain mediated repression, inhibition by ADP and, supposedly, competition by Hsp70 for binding sites on the partially disordered proteins ensures that the Hsp104 WT is tuned down to the level that is safe for the cell.

Due to its remarkable ability to dissolve protein aggregates related to stress, aging, as well as to neurodegenerative diseases (*Erjavec et al., 2007*; *Jackrel et al., 2014*; *Torrente et al., 2016*), Hsp104 and especially its derepressed variants have been the subject of intense search for their potential therapeutic application (*Castellano et al., 2015*). Our studies show that any plans for application of these protein remodeling machines in the novel environment should involve evaluation of ATP and ADP concentrations, which determine Hsp104 activity.

The knowledge of ADP influence on Hsp104 is also important for interpretation of in vitro experiments. As observed during fRCMLa proteolysis by HAP or GFP disaggregation by Hsp104 D484K, accumulating ADP inhibited the reaction before the total pool of the protein substrate has been processed. This affected the renaturation yield, a parameter that should not therefore be used to evaluate Hsp104 activity. Furthermore, the kinetics of the reaction is strongly influenced by the efficiency of different ATP regeneration systems.

In the light of Hsp104 dependence on adenine nucleotides, we integrate the established facts and reveal novel aspects of the complex interplay between the processes driving Hsp104-dependent disaggregation: the ATPase cycle, protein translocation trough the central channel, Hsp104 derepression trough the M-domain and the upstream action of Hsp70. The mechanism which emerges from our work implies that, in combination with the restrictive cellular ATP and ADP concentrations, Hsp70 provides an additional level of Hsp104 control, effectively limiting the toxic unspecific protein unfolding activity of Hsp104, at the same time supporting efficient recovery of aggregated proteins, the process crucial for cell survival after stress.

## Materials and methods

### Proteins

Mutations in *HSP104* gene were introduced using QuickChange Site Directed Mutagenesis Kit (Stratagene, USA). Published protocols were used for the purification of Hsp104 and its variants, Ydj1 (*Lipińska et al., 2013*), GFP (*Zietkiewicz et al., 2004*), Ssa1 (*Andréasson et al., 2008*) and ClpP

(*Maurizi et al., 1994*). HAP variant of Hsp104 (G739I S740G K741F T746A) was expressed from pET24a-*HAP* plasmid, a gift from Bernd Bukau (University of Heidelberg). Sis1 was purified from *E. coli* BL21 (DE3) *dnaK* carrying plasmid pET11a-*SIS1*, a gift from Elizabeth Craig (University of Wisconsin). Purification procedure involved cell lysis with French pressure cell (Thermo Scientific, USA) and Q--sepharose FF (GE-Healthcare, UK), SP-sepharose FF (Sigma-Aldrich, USA) and hydroxyapatite (BIO-RAD) column chromatography. fRCMLa was prepared from α-lactalbumine (Sigma-Aldrich, USA) according to (*Bösl et al., 2005*). Non-labelled RCMLa, α-casein, lactic dehydrogenase, pyruvate kinase and creatine kinase were purchased from Sigma-Aldrich, USA. All given protein concentrations refer to monomer.

## ATPase activity

Hsp104 WT, E285Q, E687Q or D484K were incubated at 30°C in the ATPase buffer (20 mM HEPES pH 7.5, 25 mM NaCl, 100 mM KCl, 15 mM magnesium acetate, 1 mM DTT, 10% glycerol), containing 50 μCi/ml ATPγ$^{33}$P and the indicated levels of ATP and ADP. Hsp104 concentration in each experiment was adjusted to result in hydrolysis of approximately 20% of ATP in 20 min. Initial rate of release of inorganic phosphate was assessed as described (*Viitanen et al., 1990*). Measurements of Hsp104 ATPase activity during GFP disaggregation were performed in the renaturation buffer (40 mM HEPES pH 7.5, 60 mM potassium glutamate, 15 mM magnesium acetate, 5 mM DTT). In these experiments, the reaction mixtures containing Ssa1, Sis1 and aggregated GFP were incubated for 8 min at 30°C prior to addition of Hsp104.

Steady-state ATPase assay was performed for the assessment of the *apparent* $K_M$ values for Hsp104 and Hsp104 D484K as described previously (*Nørby, 1988*). Reaction mixture containing 0.25–2 μM Hsp104, buffer (50 mM HEPES pH 7.5, 150 mM KCl, 20 mM magnesium acetate, 10 mM DTT), 0.265 mM NADH, 100 U/ml lactic dehydrogenase, 70 U/ml pyruvate kinase (LDH/PK) and 2.8 mM PEP was incubated for 5 min at 30°C. NADH absorbance was measured at 340 nm in a JASCO V-750 Spectrophotometer. *Apparent* $K_M$ values were calculated by using least squares fitting to the Michaelis–Menten equation, with the GraphPrism software. ATPase activity was calculated for Hsp104 monomer.

## ADP concentration measurements

Samples of the GFP renaturation reaction mixture (containing 1 μM Hsp104 D484K and optionally 2 μM Ssa1 and 0.5 μM Ydj1) were analyzed using reversed-phase high performance liquid chromatography (RP-HPLC) according to (*Smolenski et al., 1990*) on GBC 1150 HPLC Pump, Spectra System AS3000 autosampler, Thermo Finnigan Spectra System UV6000LP. The separation was performed on 50 x 4,6 mm HyperClone 3u BDS C18, 130 A column (Phenomenex).

### Fluorescence anisotropy

Substrate binding assay was performed as described (*Bösl et al., 2005*). Complex formation between Hsp104 E284Q and fRCMLa or between Ssa1-Ydj1 and fRCMLa was detected by following fluorescence anisotropy signal, measured in a JASCO FP-8000 Fluorescence Spectrometer. fRCMLa (1 μM) was incubated at 30°C in the renaturation buffer, containing 2.6 mM ATP and at the increasing concentrations of ADP. Subsequently, Hsp104 E285Q (12 μM) or Ssa1 (2 μM) with Ydj1 (0.5 μM) were added to the reaction mixture. After 200 s, fRCMLa release from Hsp104 was induced by addition of non-labeled RCMLa to the final concentration 40 μM.

### Proteolysis of fRCMLa

fRCMLa (1.25 μM, 5 μM or 20 μM, as indicated) was incubated for 2 min with ClpP at 30 °C and subsequently HAP was added. Reaction was carried out in the renaturation buffer, containing 2.6 mM or 10 mM ATP and, optionally, ADP or/and Ssa1 and Ydj1, as indicated. fRCMLa proteolysis by HAP-ClpP was monitored by following fluorescence anisotropy in the JASCO FP-8000 Fluorescence Spectrometer.

### Proteolysis of α-casein

α-casein (20 μM) was incubated at 30°C with HAP (1 μM), ClpP (3.6 μM). Reaction was carried out in the renaturation buffer, containing 2.6 mM or 10 mM ATP and, optionally, 1 mM ADP, 2 μM Ssa1

and 0.5 µM Ydj1 or an ATP regeneration system comprising 0.2 mg ml$^{-1}$ creatine kinase and 20 mM creatine phosphate. The samples of reaction mixtures before and after incubation were subjected to SDS-PAGE and Coomassie Brilliant Blue staining.

## Refolding of heat-aggregated GFP

GFP renaturation assay was performed as described previously (*Zietkiewicz et al., 2004*), with slight modifications. Briefly, GFP (4 mg ml$^{-1}$) was thermally inactivated at 85°C for 15 min. The reactivation reaction was carried out at 25°C in the renaturation buffer, with 10 mM or 2.6 mM ATP, and, optionally, with the indicated levels of ADP or an ATP regeneration system comprising 0.2 mg ml$^{-1}$ pyruvate kinase and 40 mM phosphoenolpyruvate or 0.2 mg ml$^{-1}$ creatine kinase and 120 mM creatine phosphate. Renaturation was initiated by adding the indicated chaperone proteins: Hsp104 WT (1 µM), D484K (0.06–1 µM, as indicated), Ssa1 (2 µM), Ydj1 (0.5 µM) or Sis1 (0.4 µM). Sis1 was used instead of Ydj1 for reactions requiring longer incubation times, due to observed low stability of Ydj1. GFP fluorescence was detected in JASCO FP-8000 Fluorescence Spectrometer or a Beckman Coulter DTX880 microplate reader.

## Acknowledgements

This work was supported by a grant of the Polish National Science Centre (2013/08/A/NZ1/00683). AK is a recipient of grant from the Ministry of Science and Higher Education "Diamentowy Grant "(0214/DIA/2012/41). We thank Drs. Jarosław Marszalek and Anna Badowiec for discussions. We thank Drs. Adrianna Radulska, Ewa Słonimska and Ryszard T Smoleński for their help in ADP concentration measurements by HPLC analysis. We thank Drs. Bernd Bukau and Elizabeth Craig for the plasmids used in the study.

## Additional information

### Funding

| Funder | Grant reference number | Author |
| --- | --- | --- |
| Narodowe Centrum Nauki | 2013/08/A/NZ1/00683 | Krzysztof Liberek |
| Polish Ministry of Science and Higher Education | 0214/DIA/2012/41 | Agnieszka Kłosowska |

The funders had no role in study design, data collection and interpretation, or the decision to submit the work for publication.

### Author contributions

AK, Conception and design, Acquisition of data, Analysis and interpretation of data, Drafting or revising the article; TC, Acquisition of data, Analysis and interpretation of data; KL, Conception and design, Analysis and interpretation of data, Drafting or revising the article

### Author ORCIDs

Krzysztof Liberek, http://orcid.org/0000-0002-7532-9279

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
