## [Decision Letter]

Thank you for submitting your article "Inhibition by ADP restricts the protein remodeling activity of Hsp104 chaperone to Hsp70 assisted disaggregation" for consideration by *eLife*. Your article has been reviewed by 3 peer reviewers, one of whom is a member of our Board of Reviewing Editors, and the evaluation has been overseen by John Kuriyan as the Senior Editor.

The reviewers have discussed your paper and considered each other's comments and the Reviewing Editor has drafted this decision to help you prepare a revised submission.

Summary:

All three reviewers recognized the significance of your findings and the elegance of the model you propose.

The main critiques to emerge from the review process concern the strength of the correlation you have established between (A) The ability of substrate – introduced by the Hsp70/40 pair – to stimulate ATP binding (and the associated allosteric changes and ATPase activity) and (B) the observed rate of substrate translocation through Hsp104. And the robustness of this correlation across a range of ATP/ADP concentrations. This point is crucial especially given the comments of reviewer three on the two papers cited in reference for the choice of the canonical concentration of ATP and ADP in the yeast cytosol: As the concern regarding the validity of the choice of nucleotide concentration would be mitigated if the aforementioned correlations were documented over a broad range of ATP/ADP concentrations.

A specific weakness were the differences in conditions under which the ATPase assays and substrate translocation assays were performed

Essential revisions:

1) Expanding Figure 2 to contain both ATPase rate determinations and disaggregation assays under identical buffer conditions (including the GFP aggregates) and the same range of ADP concentrations would be a valuable element in the study. (Or put differently, redoing the ATPase rate determinations in Figure 1 under conditions identical to the conditions in Figure 2.) This would, for example, allow the authors to plot the ATPase rates against the disaggregation rates over a range of ADP concentrations to visualize the interaction of Hsp70/40 with the ADP-inhibition.

If possible these ADP titrations could be carried out at different ATP concentrations (e.g. 2.6 mM and 10 mM) but we leave this last point to the author's discretion.

2) HAP-D484K seems less sensitive to ATP/ADP mixtures (Figure 5—figure supplement 3) as compared to HAP-WT (Figure 5) (compare decrease of fRCLMa anisotropy curves). HAP-D484K in presence of ADP seems as active as HAP-WT in absence of ADP. This conflicts with the statement that hyperactive HAP-D484K is as sensitive as HAP-WT towards ADP inhibition. Processing of fRCLMa by HAP-D484K is almost complete after 5 min incubation; might this be the reason that no inhibition was observed. The authors should clarify this point and revise their Discussion accordingly.

Discretional points to take account of:

1) Please provide further information validating your choice of ATP/ADP concentrations in the yeast cytosol addressing the concerns of reviewer 3 in regards to Ytting et al. 2012 and Özalp et al. 2010 as well as the potential role of adenylated kinase in converting ADP to AMP and ATP, as articulated by the comments of reviewer 1.

2) Binding of RCLMa to Hsp104 is suggested to increase affinity for ATP (Figure 6), thereby overcoming ADP inhibition in the end (explaining ADP-resistance of Hsp104 in presence of Hsp70, which targets Hsp104 to aggregates). However, ADP is strongly inhibiting RCLMa processing by the HAP variant (Figure 1) at the same time. Since HAP binds and degrades RCLMa (together with ClpP) one would expect resistance towards ADP, which is however not the case. Could the RCLMa concentration used here be too low to affect the affinity of HAP for ATP (as ATPase assays were done under much higher, saturating RCLMa concentrations). Please discuss these issues in an effort to reconcile the divergent findings.

3) Please review the figure legends with an eye to rendering them interpretable independently of the text. The sophisticated reader should be able to grasp the essence of the paper simply by visiting the figures and figure legends.

Reviewer #1:

This paper is concerned with the mechanism of protein de-aggregation by the yeast chaperone Hsp104. It sheds light on the dependence of de-aggregation by Hsp104 on the collaboration of an Hsp70-Hsp40 pair. The key findings concern a feature of Hsp104, namely its low affinity for ATP and its susceptibility to inhibition by ADP.

In what appear to be a series of well done in vitro experiments, the authors confirm the relative inactivity of Hsp104 at (what they claim are) physiological ADP/ATP concentrations and the marked stimulatory activity of the Hsp70-Hsp40 pair at these physiological circumstances. Using a mutant Hsp104 that is frozen in the active allosteric state (normally induced by Hsp70) they go on to show that this stimulatory effect is *not* due to allosteric activation of Hsp104 by Hsp70-Hsp40 but rather by their ability to "feed" substrate proteins to the Hsp104 machine. These last observations are rendered interpretable by the finding that substrate proteins stimulate the ATPase activity of Hsp104 and overcome the inhibitory effect of ADP. A final physiological touch is added by the observation that the stimulatory effect of Hsp70-Hsp40 of the reluctant (ADP-inhibited) Hsp104 is limited to aggregated proteins and does not extend to natively disordered protein. The story hangs together well.

I am not in a position to judge the level of novelty and there is a concern that the central conclusion – that these mechanistic features evolved to entrain Hsp104 to selectively de-aggregate and not to interfere with natively-disordered proteins – rests on too narrow a comparison of model substrates (GFP, fRCMLA). A mutant Hsp104 that is not inhibited by ADP would be extremely useful to test the authors' hypothesis, but I assume one does not exist.

Finally, despite references to the contrary (duly provided) I am not sure about the relevance of millimolar concentrations of ADP in yeast: I thought ADP gets quickly converted to ATP and AMP by adenylate kinase, an enzyme that is also present in yeast, with the effect that ADP concentration are kept very low.

Reviewer #2:

Hsp104 cooperates with the Hsp70 chaperone system in the reactivation of aggregated proteins. Hsp70 recruits Hsp104 to protein aggregates by interacting with Hsp104 M-domains, thereby liberating Hsp104 ATPase activity from M-domain repression. Derepressed Hsp104 M-domain mutant variants exhibit unleashed ATPase activity and become partially independent from Hsp70. Here, the authors show that physiological ATP/ADP ratios fine-tune Hsp70-Hsp104 cooperation by making Hsp104 activity more dependent on the Hsp70 partner. It is demonstrated that accumulation of ADP strongly poisons Hsp104 ATPase and threading activity in absence of Hsp70. This can be explained by previous findings showing higher affinity of Hsp104 for ADP as compared to ATP. Remarkably, the inhibitory effect of ADP is overcome in presence of Hsp70, ensuring that Hsp104 activity is restricted to Hsp70-coated protein aggregates. The authors provide evidence that substrate binding and translocation increases Hsp104 affinity for ATP, overruling ADP inhibition. The substrate targeting function of Hsp70 is in line with the presented model.

The presented findings are novel and interesting and provide valuable detailed insights into Hsp104 regulation and the mechanism of protein disaggregation. How exactly ADP inhibition is overcome upon substrate binding remains speculative. The authors suggest that ADP release from Hsp104 NBD2 is triggered upon substrate interaction. Direct evidence for this scenario is missing but is also not considered to be mandatory for acceptance of the manuscript. The following aspects should be addressed in a revised manuscript:

1) The authors frequently use varying nucleotide concentrations and ATP/ADP ratios. It is advised to perform all analysis (e.g. entire Figure 3) at the physiological ATP/ADP ratio (2,6 mM ATP / 1 mM ADP) allowing for better comparison and strengthening the physiological relevance of the acquired data.

2) The authors show that the apparent affinity for ATP is reduced in presence of substrate fRCLMa(Figure X). On the other hand, the processing of exactly the same substrate is strongly inhibited by ADP. Can the authors provide a rationale for these seemingly conflicting data? Is the fRCLMa concentration used in the degradation assays simply too low?

3) A derepressed Hsp104 variant seems less sensitive to ADP inhibition (e.g. efficient processing of fRCLMa in presence of ATP/ADP mixtures: Figure 5—figure supplement 3). Is it possible that the conformational state of M-domains (repressed or derepressed) additionally controls nucleotide affinities? The observation is also not in line with the statement that derepressed Hsp104 variants are as sensitive as Hsp104 wild type towards ADP inhibition.

Reviewer #3:

The manuscript by Klosowska, Chamera and Liberek describes a mechanism that restricts the yeast Hsp104 protein unfolding activity and provides substrate specificity by employing the chaperones Hsp70-Hsp40.

In summary, the authors build on the earlier findings that tight ADP binding inhibits Hsp104 ATPase activity and that ADP levels in the yeast cytoplasm reaches levels as high as 1 mM. Using a reconstituted system the authors show that Hsp104 ATPase and unfolded-protein translocation activities are limited by near physiological levels of ADP (94% and 95% inhibition at 1 mM ADP, respectively). In contrast ADP only inhibits 50% of the Hsp104-mediated disaggregation activity in the presence of Hsp70-Hsp40 at 1 mM ADP. The relative insensitivity of the disaggregation activity to ADP inhibition is explained by increased affinity for ATP during substrate translocation. The authors propose that ADP inhibits Hsp104 in the cell, and that Hsp70-Hsp40 promotes binding of Hsp104 to aggregates resulting in translocation that activates the ATPase activity.

This manuscript provides a series of elegant in vitro experiments and a simple attractive model that explains how Hsp104 activity is regulated in the cell. The manuscript is well written, easy to understand and addresses an important biological mechanism. A complementary set of in vivo experiments to test the proposed model would make the study more conclusive.

1) Relevance of Hsp104-regulation by ADP inhibition depends on a specific in vitro 'physiological condition'

A concern that has to be raised is that the observed regulation of Hsp104 activity does not reflect in vivo behavior of the protein. This concern becomes important in context of this study since the experimental setup contrasts the traditional biochemical approach when studying enzymes by not employing conditions under which Hsp104 displays optimal activity (For Hsp104: high ATP and low ADP). It is under such non-optimal buffer conditions ('physiological condition'; 1 mM ADP and 2.6 mM ATP) that Hsp104 displays the potentially physiologically relevant behavior of activity-assay specific ADP inhibition; lower ADP inhibition of the Hsp70-Hsp40-dependent disaggregation activity than of the substrate-free ATPase and translocation activities. Thus, a concern is that the in vitro assay conditions are not a good proxy for the in vivo conditions and suffer from problems with biological relevance.

1.1) Literature problem

The authors refer to Ytting et al. 2012 and Özalp et al. 2010 to argue that the ADP concentration in yeast cells oscillates around 1 mM (row 95). However, Ytting et al. 2012 does not report on ADP levels and Özalp et al. 2010 reports on ATP measurements and calculated ADP levels _assuming_ a total adenine nucleotide pool of 3.6 mM. Thus, the notion that the used nucleotide levels mimic physiological conditions is not well supported by the cited papers.

1.2) Robustness of the model

A key question is whether the ADP inhibition of Hsp104 is apparent under various buffer conditions (ADP-ATP ratios and total nucleotide concentration). If the Hsp104 behavior is robust under many different buffer conditions, it is likely to represent a mechanism of physiological importance. If the window of function for their proposed mechanism is narrow, it is likely not of physiological importance. Can the authors experimentally define the window of ADP-ATP ratios and total nucleotide concentrations where their proposed ADP-inhibition mechanism would be functional?

2) Targeting Hsp104 to aggregated substrates by Hsp70

The authors propose that Hsp70 not only activates Hsp104 by repositioning the M-domain but also relieves ADP-inhibition by physically linking Hsp104 to translocation competent substrates (for example see paragraph starting at row 364). This important part of their model is readily testable by linking Hsp104 or its M-domain repositioned variant D484K to the aggregates using fusions to dimerization domains (e.g. leucine zippers, interacting charged α-helices, etc.). Their model predicts that merely bringing Hsp104 to the vicinity of aggregates should result in acceleration of disaggregation and ATPase activity in the presence of ADP. Such as direct test of the proposition that Hsp70-independent physical targeting regulates the ADP-inhibited ATPase rates potentially strengthens their model.

---

## [Author Response]

Essential revisions:

1) Expanding Figure 2 to contain both ATPase rate determinations and disaggregation assays under identical buffer conditions (including the GFP aggregates) and the same range of ADP concentrations would be a valuable element in the study. (Or put differently, redoing the ATPase rate determinations in Figure 1 under conditions identical to the conditions in Figure 2.) This would, for example, allow the authors to plot the ATPase rates against the disaggregation rates over a range of ADP concentrations to visualize the interaction of Hsp70/40 with the ADP-inhibition.

*If possible these ADP titrations could be carried out at different ATP concentrations (e.g. 2.6 mM and 10 mM) but we leave this last point to the author's discretion*.

The suggested experiment was performed and included in the manuscript (Figure 2 and in the second paragraph of the subsection “Hsp70 allows Hsp104 to overcome ADP inhibition”). Our results show that during disaggregation performed in the presence of Hsp70 the ATPase activity of Hsp104 was much less sensitive to ADP comparing to the control, where the ATPase activity of Hsp104 alone was assessed in the same buffer and nucleotide concentrations. Additionally, we measured the ATPase activity of Hsp70 (with Hsp40 and GFP aggregates), which allows to estimate the ATPase activity attributed to Hsp104 in the mixture of Hsp104 and Hsp70 proteins.

The degree of inhibition of the ATPase activity (Figure 2) and of the disaggregation activity (Figure 2) are in good agreement, showing only 40-50% reduction at 2.6 mM ATP and 1 mM ADP, supporting our observation that Hsp70 allows Hsp104 to overcome inhibition by ADP.

2) HAP-D484K seems less sensitive to ATP/ADP mixtures (Figure 5—figure supplement 3) as compared to HAP-WT (Figure 5) (compare decrease of fRCLMa anisotropy curves). HAP-D484K in presence of ADP seems as active as HAP-WT in absence of ADP. This conflicts with the statement that hyperactive HAP-D484K is as sensitive as HAP-WT towards ADP inhibition. Processing of fRCLMa by HAP-D484K is almost complete after 5 min incubation; might this be the reason that no inhibition was observed. The authors should clarify this point and revise their Discussion accordingly.

The experiment showing the influence of ADP on degradation of fRCMLa by HAP D484K and ClpP is now presented (Figure 3—figure supplement 2 and in the fourth paragraph of the subsection “Hsp70 allows Hsp104 to overcome ADP inhibition”). Data directly show that addition of ADP substantially inhibits the ability of HAP D484K to translocate fRCMla. The degree of inhibition by ADP is comparable for the derepressed HAP D484K and the repressed HAP (Figure 1—figure supplement 1, Figure 3—figure supplement 2). At the same time, the absolute activity is much higher for HAP D484K. This is why upon ADP inhibition the activity of this variant is similar to the non-inhibited HAP.

These results are consistent with the observed effects of ADP on ATP hydrolysis and on the disaggregation activity displayed by Hsp104 D484K alone. Hsp104 D484K did not show higher affinity towards ATP (in the third paragraph of the aforementioned subsection), its relative inhibition by ADP was similar to the WT (Figure 1, Figure 3 and Figure 3—figure supplement 1) and Hsp104 D484K was almost inactive in GFP disaggregation without Hsp70 at the ATP:ADP ratio below 5:1 (Figure 3—figure supplement 3). We feel it is justified to conclude from these results that conformation change of the M-domain does not release ADP inhibition and is not enough to enable efficient disaggregation in the presence of ADP.

At the same time, we do not exclude the possibility that stimulation of Hsp104 activity associated with the interaction between Hsp70 and the M-domain might facilitate Hsp70-dependent binding to aggregates, which has been stated in the Discussion (eighth paragraph).

Discretional points to take account of:

1) Please provide further information validating your choice of ATP/ADP concentrations in the yeast cytosol addressing the concerns of reviewer 3 in regards to Ytting et al. 2012 and Özalp et al. 2010 as well as the potential role of adenylated kinase in converting ADP to AMP and ATP, as articulated by the comments of reviewer 1.

Several studies (Hynne et al., 2001, Biophysical Chemistry, 94, 121; Teusink et al., 2000, Eur. J. Biochem. 267, 5313; Wu et al., 2006, Appl. Environ. Microbiol., 72, 3566; references added to reference list), including our own measurements (Figure 8) show that ADP is at significant proportion to ATP in yeast cells (Table 1 below). As summarized in the report by Canelas et al., 2008, Biotechnol. Bioeng. 100, 734; added to reference list), the range of the cellular concentrations of ATP is approximately 2.1-3.4 mM and of ADP: 0.5 mM -1.5 mM. We noted that ATP and ADP concentrations assessed in intact cells (Ozalp et al. 2010) (2.6 mM of ATP and 1 mM of ADP) are average values of the range of concentrations presented in the previous reports (Canelas et al., 2008). This inspired us to use the latter ATP/ADP levels as a representation of physiological concentrations of adenine nucleotides in the in vitro analyses of Hsp104 activities.

Author response image 1.ATP:ADP ratio in yeast cell extracts.Yeast cells (diploid strain X2180) were grown at 30°C to 1.4 * 10^8^ cells ml^-1^ in YPD (**A**) or 0.7 * 10^8^ cells ml^-1^ in YPGal medium (**B**). Cells were quickly harvested through centrifugation (5000 g, 1’), washed twice in distilled water and resuspended in 0.2 M perchloric acid (400 μl per 100 mg of wet weight). Cell lysis was performed by two series of freezing/thawing and the extracts were neutralized by adding potassium phosphate (3M) to the final pH 7. Extracts were centrifuged twice (10 000 g, 10’) and supernatant was analyzed using reversed-phase high performance liquid chromatography (RP-HPLC) according to (Smolenski et al, 1990) on GBC 1150 HPLC Pump, Spectra System AS3000 autosampler, Thermo Finnigan Spectra System UV6000LP. The separation was performed on 50 x 4,6 mm HyperClone 3u BDS C18, 130 A column (Phenomenex).**DOI:**
http://dx.doi.org/10.7554/eLife.15159.020

Author response table 1.ATP and ADP proportions in yeast.**DOI:**
http://dx.doi.org/10.7554/eLife.15159.021ReferenceATP:ADP ratioATP (mM)ADP (mM)Growth conditionsTheobald et al., 1997*7.1:13.360.47Glucose-limitedTeusink et al., 2000*1.9:12.521.32Glucose excessHynne et al., 2001*1.4:12.11.5Glucose excessVisser et al., 2004*3.7:12.650.72Glucose-limitedOsorio et al., 20035.9:11.07 ± 0.080.18 ± 0.02Glucose excessOsorio et al., 20032.9:11.51 ± 0.320.52 ± 0.18Galactose excessOzalp et al, 20102.6:12.6^‡^1^‡^Glucose-limitedThis study^†^3.9:1 - 4.8:1––Glucose excessThis study^†^3.5:1 - 3.8:1––Galactose excess^*^Adopted from Canelas et al. (2008).^†^Each measurement was performed in two biological repeats (and samples from each biological repeat were analyzed twice). Representative result is shown in Author response image 1.^‡^ Estimated, assuming total ATP and ADP concentration of 3.6 mM.

We agree with the reviewers’ and editors’ comments that our choice of ATP/ADP concentrations was not explained enough in the previous version of the manuscript. In the revised version we included more comprehensive justification (Introduction, sixth paragraph; subsection “ADP inhibits the ATPase activity and polypeptide processing by Hsp104”, second paragraph; Discussion, second paragraph). We also stressed that the actual “physiological ATP/ADP concentrations” vary depending on cell metabolism (subsection “ADP inhibits the ATPase activity and polypeptide processing by Hsp104”). For that reason and according to the reviewers’ and editors’ suggestions, we assessed the degree of Hsp104 inhibition at a broad range of ADP concentrations (Figure 1 and Figure 2, essential revision comment 1). Furthermore, our experiments at different ATP levels (10 mM and 2.6 mM) show similar degree of inhibition for similar ATP:ADP ratios (Figure 1).

Therefore, we believe that our experimental conditions allow to estimate the degree of Hsp104 inhibition under the range of ATP/ADP concentrations that occur in yeast cells.

From the analyses of ADP level in yeast cells we conclude that the rate of ADP conversion to AMP or ATP (by Adenylate Kinase) is low enough to allow for residual ADP level to be at the millimolar range.

2) Binding of RCLMa to Hsp104 is suggested to increase affinity for ATP (Figure 6), thereby overcoming ADP inhibition in the end (explaining ADP-resistance of Hsp104 in presence of Hsp70, which targets Hsp104 to aggregates). However, ADP is strongly inhibiting RCLMa processing by the HAP variant (Figure 1) at the same time. Since HAP binds and degrades RCLMa (together with ClpP) one would expect resistance towards ADP, which is however not the case. Could the RCLMa concentration used here be too low to affect the affinity of HAP for ATP (as ATPase assays were done under much higher, saturating RCLMa concentrations). Please discuss these issues in an effort to reconcile the divergent findings.

In the response to the reviewers’ and editors’ comments we assessed the degree of ADP inhibition of fRCMLa proteolysis by HAP-ClpP at various concentrations of fRCMLa (Figure 1—figure supplement 1). The results show that the absolute rate of substrate degradation increases with the substrate level, however the relative inhibition by ADP is similar for all the analyzed fRCMLa concentrations.

This observation be explained by dividing fRCMLa processing by Hsp104 into two phases: protein binding and translocation. Protein binding is highly affected by ADP (Figure 1) and protein translocation is associated with increased affinity to ATP and partial resistance to ADP (Figure 6). Our result suggests that translocation of one protein substrate has to be finished before another substrate is able to bind to HAP, therefore the ATPase stimulation associated with protein translocation does not promote substrate binding.

We speculated that with decreasing ATP:ADP ratio the frequency of occurrence of Hsp104 conformations allowing for substrate binding decreases. When binding to Hsp104 is available, increasing fRCMLa concentration increases the probability of its successful binding to Hsp104 (or HAP), resulting in higher observed rate of fRCMLa proteolysis. However, substrate concentration does not seem to affect, how often Hsp104 is in a binding-competent state, the latter being determined by ATP:ADP ratio. This could explain, why the relative effect of ADP is similar regardless of fRCMLa concentration. We have revised our Discussion in this context (sixth paragraph).

3) Please review the figure legends with an eye to rendering them interpretable independently of the text. The sophisticated reader should be able to grasp the essence of the paper simply by visiting the figures and figure legends.

To improve the clarity of the presentation of our results, we added experimental scheme in the Figure 2. Also, we changed design of Figure 1 to show results obtained under analogous conditions, excluding order of addition experiments. Several corrections were also introduced into Figure legends.

Reviewer #1:

*This paper is concerned with the mechanism of protein de-aggregation by the yeast chaperone Hsp104. It sheds light on the dependence of de-aggregation by Hsp104 on the collaboration of an Hsp70-Hsp40 pair. The key findings concern a feature of Hsp104, namely its low affinity for ATP and its susceptibility to inhibition by ADP.*

*In what appear to be a series of well done* in vitro *experiments, the authors confirm the relative inactivity of Hsp104 at (what they claim are) physiological ADP/ATP concentrations and the marked stimulatory activity of the Hsp70-Hsp40 pair at these physiological circumstances. Using a mutant Hsp104 that is frozen in the active allosteric state (normally induced by Hsp70) they go on to show that this stimulatory effect is not due to allosteric activation of Hsp104 by Hsp70-Hsp40 but rather by their ability to "feed" substrate proteins to the Hsp104 machine. These last observations are rendered interpretable by the finding that substrate proteins stimulate the ATPase activity of Hsp104 and overcome the inhibitory effect of ADP. A final physiological touch is added by the observation that the stimulatory effect of Hsp70-Hsp40 of the reluctant (ADP-inhibited) Hsp104 is limited to aggregated proteins and does not extend to natively disordered protein. The story hangs together well.*

I am not in a position to judge the level of novelty and there is a concern that the central conclusion – that these mechanistic features evolved to entrain Hsp104 to selectively de-aggregate and not to interfere with natively-disordered proteins – rests on too narrow a comparison of model substrates (GFP, fRCMLA). A mutant Hsp104 that is not inhibited by ADP would be extremely useful to test the authors' hypothesis, but I assume one does not exist.

Finally, despite references to the contrary (duly provided) I am not sure about the relevance of millimolar concentrations of ADP in yeast: I thought ADP gets quickly converted to ATP and AMP by adenylate kinase, an enzyme that is also present in yeast, with the effect that ADP concentration are kept very low.

Please see our response to the editors’ discretional point 1, Figure 8 and Table 1.

Reviewer #2:

The presented findings are novel and interesting and provide valuable detailed insights into Hsp104 regulation and the mechanism of protein disaggregation. How exactly ADP inhibition is overcome upon substrate binding remains speculative. The authors suggest that ADP release from Hsp104 NBD2 is triggered upon substrate interaction. Direct evidence for this scenario is missing but is also not considered to be mandatory for acceptance of the manuscript. The following aspects should be addressed in a revised manuscript:

1) The authors frequently use varying nucleotide concentrations and ATP/ADP ratios. It is advised to perform all analysis (e.g. entire Figure 3) at the physiological ATP/ADP ratio (2,6 mM ATP / 1 mM ADP) allowing for better comparison and strengthening the physiological relevance of the acquired data.

As suggested by the reviewer, we performed analysis of Hsp104 D484K ATPase activity at the physiological ATP (2.6 mM) and ADP (1 mM) concentrations (Figure 3—figure supplement 1). The degree of inhibition is similar as observed for the WT Hsp104, although the absolute activity was much higher for Hsp104 D484K, which is in agreement with the elevated protein translocation activity (Figure 5—figure supplement 3).

We also measured ATP hydrolysis by Hsp104 D484K at 10 mM ATP and at increasing concentrations of ADP (Figure 3), analogously as for the WT (Figure 1). We believe that now the results provide good comparison between the activity of Hsp104 WT and its derepressed variant D484K.

The disaggregation activity of Hsp104 D484K in the absence of Hsp70 was at a very low level and the use of 10 mM concentration of ATP in the reaction mixtures allowed for higher rate of GFP renaturation. The resulting better signal to noise ratio enabled us to detect linear increase in GFP fluorescence even at very low GFP concentrations and assess precisely the *apparent K_M_* values shown in Figure 4. At the same time, Figure 3 were designed to show only qualitative assessment of Hsp104 D484K susceptibility to ADP and the counteracting effect of Hsp70. To make the experiments presented in Figure 3 and Figure 4 comparable, we used 10 mM ATP in all of them.

2) The authors show that the apparent affinity for ATP is reduced in presence of substrate fRCLMa(Figure X). On the other hand, the processing of exactly the same substrate is strongly inhibited by ADP. Can the authors provide a rationale for these seemingly conflicting data? Is the fRCLMa concentration used in the degradation assays simply too low?

Please see our response to the editors’ discretional revision point 2.

3) A derepressed Hsp104 variant seems less sensitive to ADP inhibition (e.g. efficient processing of fRCLMa in presence of ATP/ADP mixtures: Figure 5—figure supplement 3). Is it possible that the conformational state of M-domains (repressed or derepressed) additionally controls nucleotide affinities? The observation is also not in line with the statement that derepressed Hsp104 variants are as sensitive as Hsp104 wild type towards ADP inhibition.

Please see our response to the editors’ essential revision comment 2.

Reviewer #3:

*This manuscript provides a series of elegant in vitro experiments and a simple attractive model that explains how Hsp104 activity is regulated in the cell. The manuscript is well written, easy to understand and addresses an important biological mechanism. A complementary set of in vivo experiments to test the proposed model would make the study more conclusive.*

1) Relevance of Hsp104-regulation by ADP inhibition depends on a specific in vitro 'physiological condition'

A concern that has to be raised is that the observed regulation of Hsp104 activity does not reflect in vivo behavior of the protein. This concern becomes important in context of this study since the experimental setup contrasts the traditional biochemical approach when studying enzymes by not employing conditions under which Hsp104 displays optimal activity (For Hsp104: high ATP and low ADP). It is under such non-optimal buffer conditions ('physiological condition'; 1 mM ADP and 2.6 mM ATP) that Hsp104 displays the potentially physiologically relevant behavior of activity-assay specific ADP inhibition; lower ADP inhibition of the Hsp70-Hsp40-dependent disaggregation activity than of the substrate-free ATPase and translocation activities. Thus, a concern is that the in vitro assay conditions are not a good proxy for the in vivo conditions and suffer from problems with biological relevance.

1.1) Literature problem

The authors refer to Ytting et al. 2012 and Özalp et al. 2010 to argue that the ADP concentration in yeast cells oscillates around 1 mM (row 95). However, Ytting et al. 2012 does not report on ADP levels and Özalp et al. 2010 reports on ATP measurements and calculated ADP levels _assuming_ a total adenine nucleotide pool of 3.6 mM. Thus, the notion that the used nucleotide levels mimic physiological conditions is not well supported by the cited papers.

Please see our response to the editors’ discretional point 1. Reference Ytting et al. (2012) was removed from the manuscript in response to reviewer’s comment.

1.2) Robustness of the model

A key question is whether the ADP inhibition of Hsp104 is apparent under various buffer conditions (ADP-ATP ratios and total nucleotide concentration). If the Hsp104 behavior is robust under many different buffer conditions, it is likely to represent a mechanism of physiological importance. If the window of function for their proposed mechanism is narrow, it is likely not of physiological importance. Can the authors experimentally define the window of ADP-ATP ratios and total nucleotide concentrations where their proposed ADP-inhibition mechanism would be functional?

The revised figures Figure 1, and Figure 2 allow to compare Hsp104 activities under a broad range of ATP:ADP ratios. If we define a substantial inhibition as a 50% reduction of both the ATPase activity and the protein processing activity, then for Hsp104 alone the ATP:ADP ratio at which substantial inhibition occurs is 10:1 (Figure 1). For Hsp104 in the presence of the full disaggregation system (Hsp104-Hsp70-Hsp40) substantial inhibition is observed below ATP:ADP ratio 2.6:1 for the disaggregation activity or at 3.5:1 for the ATPase activity. If we take into consideration a 90% reduction in activity as a very strong inhibition, such effect for Hsp104 alone was observed below ATP:ADP ratio 2.6:1 and for Hsp104 assisted by Hsp70 in disaggregation – below 1.3:1. Thus, the relative Hsp104 activity is maintained above 50% only by the assistance of Hsp70 across ATP:ADP ratios 10:1-2.6:1 and above 10% – across ATP:ADP ratios from 2.6:1 to 1.3:1.

For comparison, ATP:ADP ratios reported in previous studies and in this study (Figure 8) are presented in Table 1. Since the reported ATP:ADP ratios in yeast cells range from 1.4:1 to 7.1:1, we conclude that the proposed model is valid across physiological conditions. We have revised our Discussion accordingly (second paragraph).

2) Targeting Hsp104 to aggregated substrates by Hsp70

The authors propose that Hsp70 not only activates Hsp104 by repositioning the M-domain but also relieves ADP-inhibition by physically linking Hsp104 to translocation competent substrates (for example see paragraph starting at row 364). This important part of their model is readily testable by linking Hsp104 or its M-domain repositioned variant D484K to the aggregates using fusions to dimerization domains (e.g. leucine zippers, interacting charged α-helices, etc.). Their model predicts that merely bringing Hsp104 to the vicinity of aggregates should result in acceleration of disaggregation and ATPase activity in the presence of ADP. Such as direct test of the proposition that Hsp70-independent physical targeting regulates the ADP-inhibited ATPase rates potentially strengthens their model.

We agree that it would support our project if a proposed experiment turned out to be successful. However, the approach suggested by reviewer 3 poses some serious technical problems. Firstly, to introduce a dimerization domain in a vicinity of the entrance to the central channel, N-terminal fusion (or close to the N-terminus) would be required. From our experience, fusion tags introduced close to the N-terminus of Hsp104 strongly impair the protein activity and instead C-terminal tags are used (for example: Erjavec, 2007, Coelho 2014). Secondly,if a dimerization domain was fused to a substrate protein prior to aggregation, its structure would be lost under the brutal aggregation conditions. One solution would be to carry out fusion with an unstable protein, which aggregates under conditions at which the dimerization domain is stable. Since the only labile model protein for studying disaggregation is Luciferase (standard aggregation temperature is 48°C), which requires Hsp70 for proper refolding, therefore a new assay system with other model substrate would have to be developed. Other solution would be to crosslink a dimerization domain to a pre-formed aggregate. Unfortunately, the latter approach would result in the dimerization domain placed only at the surface of an aggregate and after disaggregation of the outer layer by Hsp104 the interaction site would be lost. Due to the emerging challenges, testing our model by tethering Hsp104 to aggregates is a substantial project on its own.